# The impact of community led alternative rite of passage on eradication of female genital mutilation/cutting in Kajiado County, Kenya: A quasi-experimental study

Samuel Muhula[1]*, Anthony Mveyange[2], Samuel Oji Oti[3], Martha Bande[2], Hellen Kayiaa[4], Charles Leshore[5], David Kawai[6], Yvonne Opanga[1], Enock Marita[1], Sarah Karanja[1], Eefje Smet[7], Hilke Conradi[7]

1 Department of Monitoring Evaluation and Research, Amref Health Africa in Kenya, Nairobi, Kenya, 2 Directorate of Research and Learning, TradeMark East Africa, Nairobi, Kenya, 3 Network of Impact Evaluation Researchers in Africa, Nairobi, Kenya, 4 Independent Research Consultant, Nairobi, Kenya, 5 Department of Reproductive Maternal Child and Newborn Health, Amref Health Africa in Kenya, Nairobi, Kenya, 6 End FGM/C Center of Excellence, Amref Health Africa in Kenya, Nairobi, Kenya, 7 Amref Health Africa in Netherlands, Leiden, The Netherlands

* sam.muhula@gmail.com

**Data Availability Statement:** KDHS data used in this study is available at https://dhsprogram.com/

## Abstract

### Introduction

In Kenya, Female Genital Mutilation/Cutting (FGM/C) is highly prevalent in specific communities such as the Maasai and Somali. With the intention of curtailing FMG/C prevalence in Maasai community, Amref Health Africa, designed and implemented a novel intervention—community-led alternative rite of passage (CLARP) in Kajiado County in Kenya since 2009. The study: a) determined the impact of the CLARP model on FGM/C, child early and forced marriages (CEFM), teenage pregnancies (TP) and years of schooling among girls and b) explored the attitude, perception and practices of community stakeholders towards FGM/C.

### Methods

We utilised a mixed methods approach. A difference-in-difference approach was used to quantify the average impact of the model with Kajiado as the intervention County and Mandera, Marsabit and Wajir as control counties. The approach relied on secondary data analysis of the Kenya Demographic and Health Survey (KDHS) 2003, 2008–2009 and 2014. A qualitative approach involving focus group discussions, in-depth interviews and key informant interviews were conducted with various respondents and community stakeholders to document experiences, attitude and practices towards FGM/C.

### Results

The CLARP has contributed to: 1) decline in FGM/C prevalence, CEFM rates and TP rates among girls by 24.2% (p<0.10), 4.9% (p<0.01) and 6.3% (p<0.01) respectively. 2) increase in girls schooling years by 2.5 years (p<0.05). Perceived CLARP benefits to girls included:

Data/. All qualitative data supporting the findings of this study are available as presented in this paper.

**Funding:** The study is funded by Amref Health Africa (https://amref.org/).

**Competing interests:** The authors have declared that no competing interests exist

reduction in teenage marriages and childbirth; increased school retention and completion; teenage pregnancies reduction and decline in FGM/C prevalence. Community stakeholders in Kajiado believe that CLARP has been embraced in the community because of its impacts in the lives of its beneficiaries and their families.

## Conclusion

This study demonstrated that CLARP has been positively received by the Maasai community and has played a significant role in attenuating FGM/C, CEFM and TP in Kajiado, while contributing to increasing girls' schooling years. CLARP is replicable as it is currently being implemented in Tanzania. We recommend scaling it up for adoption by stakeholders implementing in other counties that practice FGM/C as a rite of passage in Kenya and across other sub Saharan Africa countries.

## 1. Introduction

### 1.1 Female genital mutilation/cutting

Female Genital Mutilation/Cutting (FGM/C) is a global concern, particularly in Africa and the Middle East [1, 2]. Although the prevalence has started to wane, especially in Africa, recent data reveal that FGM/C is still practised as a social norm and remains persistent and ubiquitous in some communities across countries. However, concerted international efforts and commitment to address FGM/C continue to grow. Countries have made strides in designing strategies, plans, policies and in passing laws against the practice in addition to resource mobilisation to support the efforts to eradicate FGM/C [3–5].

FGM/C is a cultural practice involving partial or complete removal and alteration/injury to the external female genitalia for non-medical reasons and it may take four key forms: partial or total removal of the clitoris (Clitoridectomy or Type I); partial or complete removal of the clitoris and labia (Excision, or Type II); narrowing of the vaginal opening (Infibulation or Type III); and other harmful practices such as incising, piercing or scraping of the genital area (Type IV) [6]. Statistics show global estimates of roughly 200 million women and girls who have undergone the FGM/C practise [7, 8]. In Sub-Sahara Africa, approximately three million girls undergo FGM/C practice every year and half of the countries that practise FGM/C cut girls before the age of 5 while in other countries cutting occurs between 5 and 14 years of age [1, 9].

Across most countries, girls and women victims have undergone either clitoridectomy or excision or both as part of cultural gender and social status identity, with traditional practitioners such as cultural elders and female circumcisers/Traditional Birth Attendants (TBAs) being the main perpetrators of the practice. Higher prevalence global patterns of FGM/C are driven by migration patterns (e.g. resulting from internal and external displacement due to civil unrest and conflicts) from one part of the world to the other [8]. Most of the girls and women who have undergone the practice live in north-eastern, eastern and western parts of Africa where the practice is rife [8].

**1.1.1 What is the status of FGM/C in Kenya?** Out of approximately 125 million girls and women who underwent FGM/C by 2013, Kenya accounted for more than 9.3 million which is about 7.4 per cent of FGM/C cases [9]. The overall prevalence declined from 31.8 per cent in 2003 to 21.6 per cent in 2014 whereas the distribution across major Kenyan regions indicates alarming FGM/C prevalence in North Eastern Kenya, with an average above 90 per cent across the three survey waves and Western Kenya recording the lowest prevalence below 5 per cent (Fig 1).

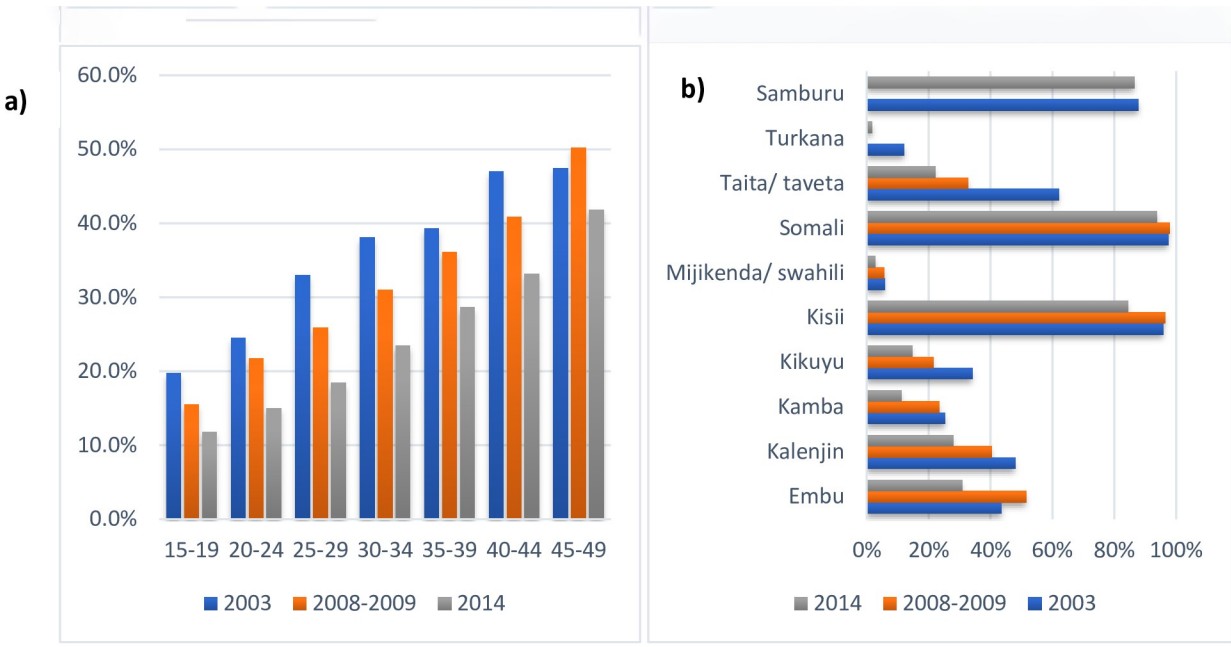

Source: Authors' construction using 2003, 2008-2009, 2014 KDHS data

**Fig 1. Graphs showing FGM/C prevalence in Kenya, 2003 to 2014.** Shows FGM/C prevalence computed using 2003, 2008–2009 and 2014 demographic health surveys data. a) Shows the overall FGM/C prevalence in Kenya from 2003 to 2014. b) shows the distribution of FGM/C prevalence by Kenyan regions.

A spatial distribution of FGM/C prevalence across Kenyan counties in 2014 shows that twenty-five counties (out of 47) had higher than average prevalence. These counties include Mandera, Wajir, Marsabit and Garissa counties where prevalence was more than 90 per cent while Kilifi, Uasin Gishu, Bungoma and Vihiga counties had low prevalence of less than 5 per cent (Fig 2).

Within Kenya, there are also wide ethnic and cultural variations in the distribution of FGM/C prevalence. Ethnic differences appear to highly correlate with high prevalence in sub-regions and counties that are predominated by the Somali, Samburu, Kisii and Maasai communities at 94, 86, 84 and 78 per cent, respectively [10, 11]. Even the specific form of FGM/C practised varies across communities. For example, 75 per cent of the FGM/C exercised by Somalis are of the most severe Type III infibulation [11, 12]. The Kisii and Maasai communities practice Type I clitoridectomy and Type II excision, respectively [11, 13]. FGM/C practise also varied significantly across different age cohorts and ethnicities over time. Fig 3 shows that not only has FGM/C been declining but it has been doing so across different age cohorts and ethnicities. Fig 3A reveals two key contradicting patterns. First, a steady decline in prevalence across years for all the age cohorts, which is consistent across the three survey rounds suggesting somewhat persistent but declining FGM/C practices. Second, disproportionate increases in prevalence across the different age cohorts in each survey year with older cohorts registering increases in the FGM/C practices. Fig 3B show declining FGM/C trends in some ethnic groups —Embu, Kalenjin, Kamba, Kikuyu, Taita, Mijikenda or Swahili on the coast, and Turkana— but reveal a higher prevalence among the Kisii, Samburu and Somali ethnic groups–with not only high FGM/C rates above 90 per cent but also the rate of reduction over the years was minuscule signalling persistence of the practices.

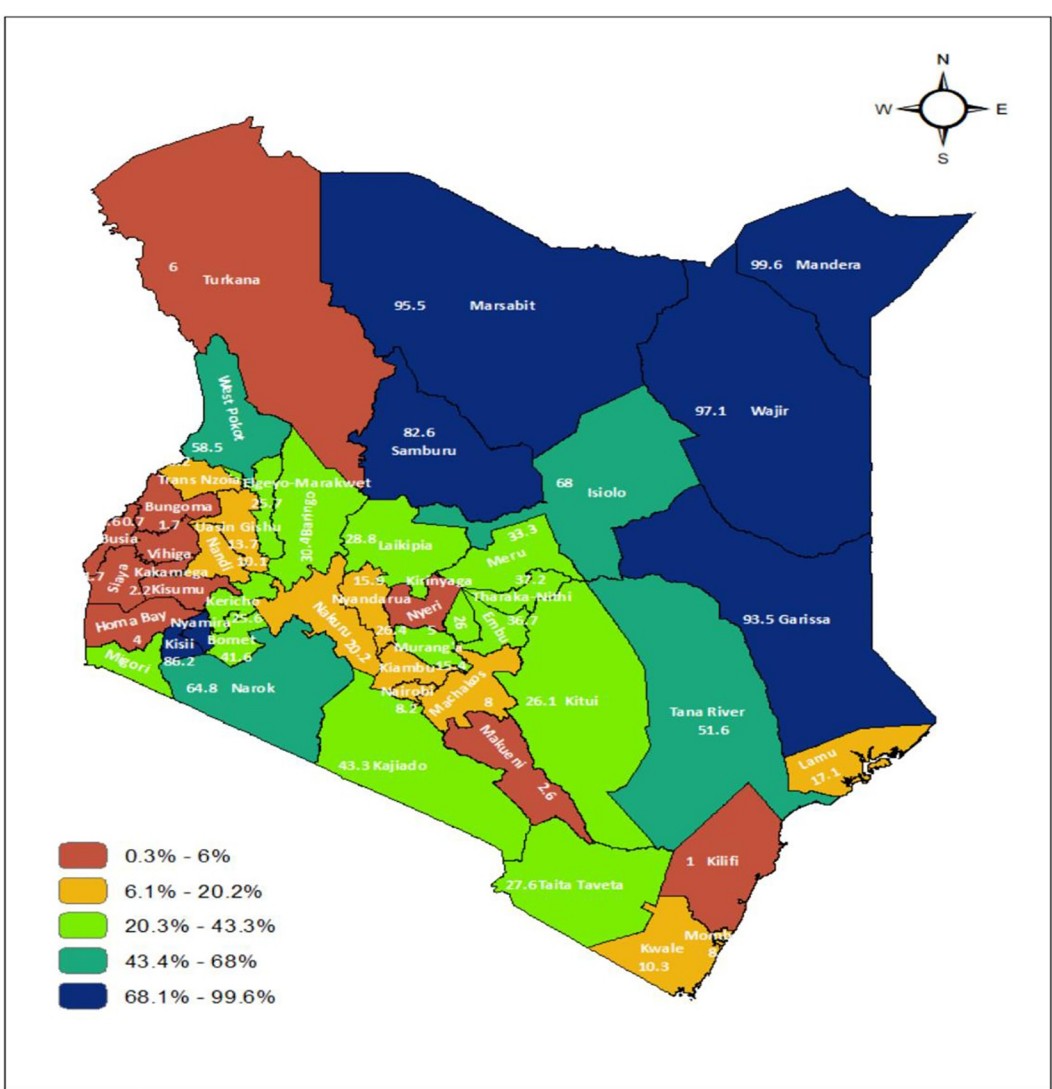

Source: Author's construction using 2014 KDHS data.

**Fig 2. A spatial distribution of FGM/C prevalence in Kenya, 2014.** Shows a spatial distribution of FGM/C prevalence in Kenya by county using the Kenya Demographic and Health Survey (KDHS) 2014 data.

CEFM, TP and FGM/C are heavily interlinked [14]. Traditional beliefs and practices are also associated with high rates of child, early and forced marriages and teenage pregnancies in Kenya. Statistics show that up to 2.3 million girls in Kenya have had their first pregnancy at adolescent years, while 535,441 women aged between 20 and 24 had their first pregnancies by the age of 18 [15]. Additionally, child marriages remain high, at 23 per cent [4]. KDHS data also tells us that 2 per cent of circumcised women age 15–49 experience FGM/C Type I, 87 per cent had Type II, and 9 per cent Type III [16]. The data further show high mother-daughter correlations—girls age 0–14 are more likely to go through the practice if their mother had undergone FGM/C–with 8 per cent of Kenyan girls age 0–14 having gone through FGM/C Type III.

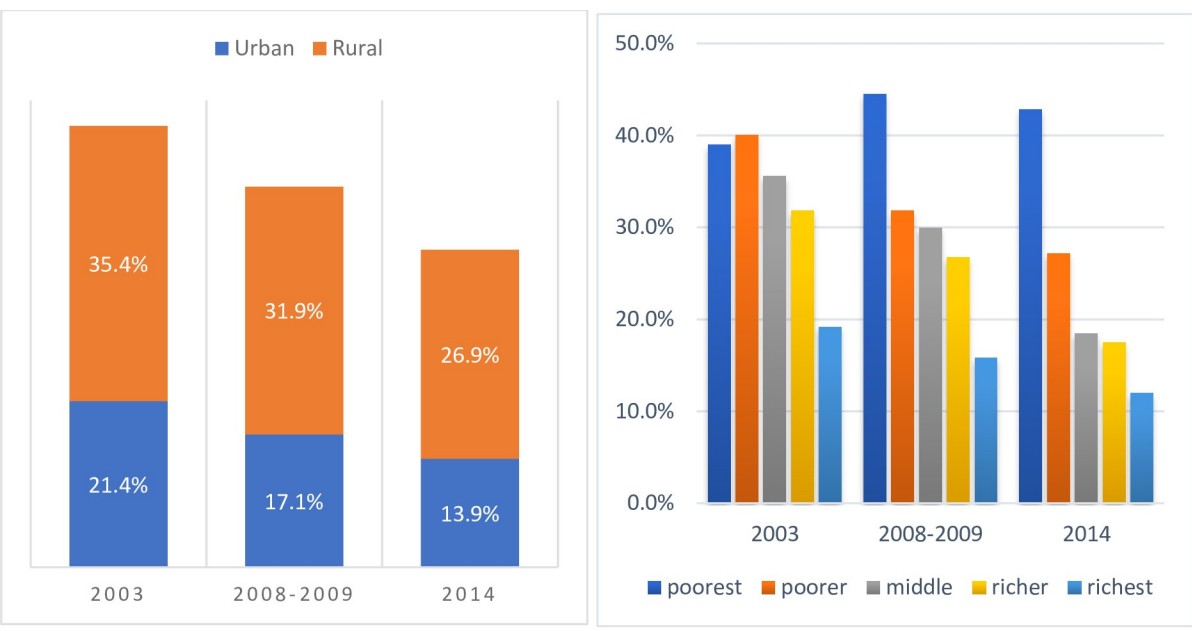

Source: Authors' construction using 2003, 2008-2009, 2014 KDHS data

**Fig 3. FGM/C prevalence across age cohorts and ethnicity.** Shows FGM/C prevalence constructed using 2003, 2008–2009 and 2014 KDHS data. a) shows FGM/C prevalence by age cohorts across the three panels of KDHS data. b) shows FGM/C prevalence by ethnic groups across the three panels of KDHS data.

Socioeconomic factors are also critical drivers of FGM/C practices in Kenya [10]. **Fig 4** indicates that prevalence were predominantly high for poor and rural girls and women. The estimates are consistent in all three survey waves across the decade reinforcing the notion that traditional beliefs and practices, especially in rural areas, exacerbated FGM/C practices in Kenya. Poverty could also somewhat explain the high prevalence, especially in rural areas as shown in Fig 4B.

Kenya has adopted several legal and legislative texts to eliminate FGM/C. Article 44 (3) of the Kenyan Constitution bars any person from compelling another person to perform, observe or undergo any harmful cultural practice or rite. Moreover, Article 53 (d) categorically states that children should be free from harmful cultural practices, inhuman and degrading treatment. However, legal actions alone are inadequate to change attitudes and behaviours as it fails to address underlying socio-cultural drivers of FGM/C [17]. Rigid enforcement of the law may result in the unintended effect of encouraging difficult to detect approaches in effecting the cut, making reporting of prevalence difficult [18]. The CLARP intervention in Kenya is one of the classic models of community-led initiatives aiming at changing perceptions, attitudes and behaviours towards FGM/C practices.

### 1.2 An overview of the CLARP model

Alternative rites of passage (ARP) is an intervention that basically allows girls to celebrate their initiation to womanhood with respect to culture and traditions without necessarily undergoing the cut [19]. In 1996, *Maendeleo ya Wanawake* movement initiated the first ARP among the Meru community who were then practising FGM as a large community celebration [3]. The aim of the ARP was to maintain the cultural practice of initiating girls to womanhood without

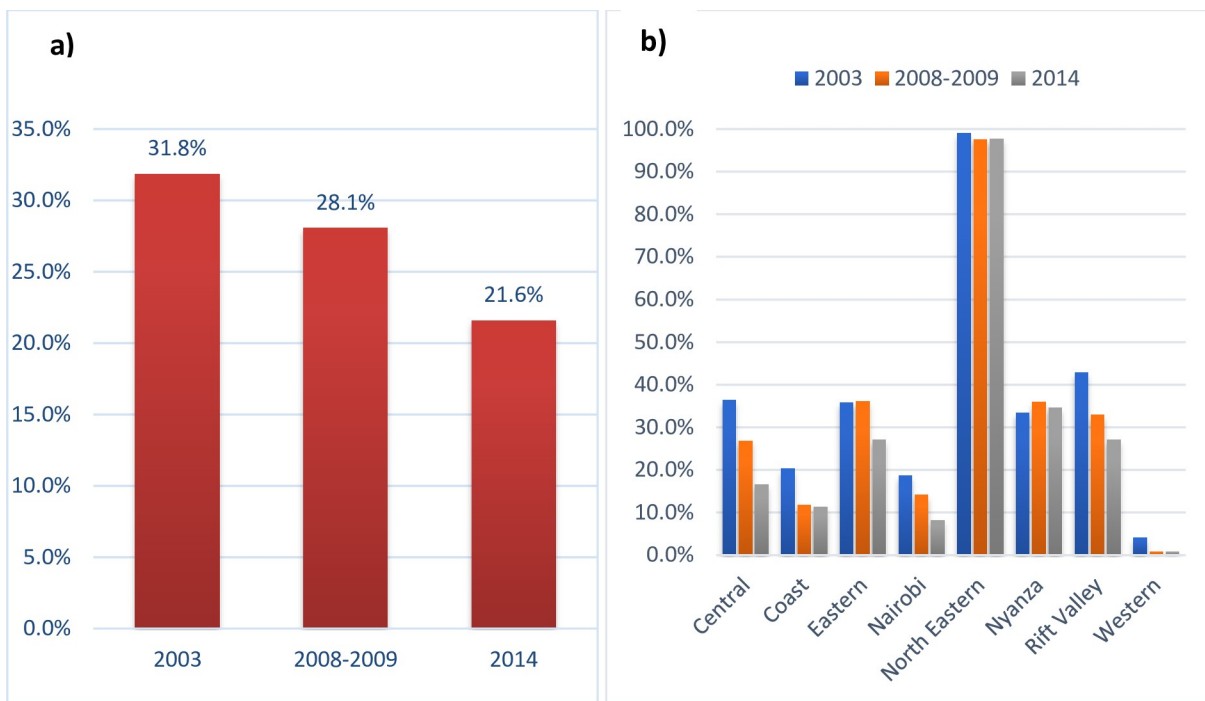

Source: Authors' construction using 2003, 2008-2009, 2014 KDHS data

**Fig 4. FGM/C prevalence by residence and wealth status.** a) shows high FGM/C prevalence for women residing in rural relative to urban areas. b) shows persistently high prevalence among poor women in all survey waves.

the cut. The intervention was successful in reducing uptake of FGM/C among girls and this led to development of different models of ARPs by NGOs in Kenya which have been implement in Kisii, Kuria and Narok [3]. A review of different ARP models indicate that for it to be taken up by communities, consideration of the sociocultural context is key [3].

In Kenya, the CLARP model was first rolled out in Kajiado County in 2009 in line with Amref's Anti-FGM/C vision to eradicate widespread FGM/C practices across the African continent by 2030 [20]. Amref Health Africa in Kenya, designed and implemented a novel intervention–dubbed Community Led Alternative Rite of Passage (CLARP) [21] with the aim of curbing the high FGM/C prevalence in the country. Successfully piloted and rolled out in Kajiado county, this community-led intervention sought to change social norms and reverse the alarming trends in FGM/C rates in Kajiado by involving and engaging community stakeholders including cultural leaders, *Morans* (young Maasai men who live in isolation in bushes, learning tribal customs and developing strengths, courage and endurance—traits for which Maasai warriors are known for throughout the world.), TBAs, County Governments, religious and cultural leaders. The whole CLARP process takes 6–48 months to complete, and it entails several steps as summarised below:

**Step 1:** Context analysis which involve engaging stakeholders such as cultural leaders, *Morans*, female circumcisers/TBA's, County government department and religious leaders.

**Step 2:** Community Led ARPs Triggering which involve structured community dialogues.

**Step 3:** Sensitization and training of cultural elders, *Morans*, women groups and circumcisers where communities define their own CLARP process.

**Step 4:** Community mobilization, sensitization & training which involve collaboration with Civil Society Organizations (CSOs) (e.g. local women groups, youth-led organisations, etc) and capacity strengthening of the CSO's through sub-grants.

**Step 5:** CLARP 3-days training of boys and girls on sexual and reproductive health rights, positive norms & values, self-esteem, life skills.

**Step 6:** CLARP ceremony with girls' graduation through CLARP, blessing by Cultural elders and leaders', Public denouncement of FGM/C.

**Step 7:** Sustaining FGM/C free communities.

The key outcomes of the CLARP interventions include the establishment of a community movement that takes action to transform social and gender norms that perpetuate FGM/C, CEFM and TPs. At the core of the CLARP, are boys and girls who are engaged and supported to know and claim their sexual reproductive health rights and take informed action when deprived of their rights. The CLARP model fosters community-led discussions and empowers not only girls and women, but also policymakers and community leaders to develop and implement local laws and policies on FGM/C and its adverse manifestations. In the end, the intervention aims to protect girls and women from FGM/C, measured by reduction in FGM/C cases, reduced cases of both CEFM and TP, and improvement in secondary school completion rates.

However, Kajiado county features prominently among counties with high prevalence for both TPs and CEFMs. CEFM rates remain high, at 28 per cent [4]. Kajiado's school enrolment rates are marginally low and stand at 75 per cent and 26.3 per cent for primary and secondary school, respectively [22]. These statistics point to a lower school enrolment rate for secondary schools, influenced by, amongst other, harmful practices such as FGM/C and CEFM. High prevalence of FGM/C cases and child marriage with its resultant effect on high teenage pregnancies in Kajiado mainly inform various interventions by national, county governments and development partners, such as Amref Health Africa. Given these higher prevalences in Kajiado, the CLARP intervention is among the essential community-led initiatives that have the potential to address FGM/C practices and the associated consequences. Therefore, CLARP model provide training targeted at reducing and eventually eliminating not only FGM/C but also CEFM and TP and improve girls' or women's levels of educational outcomes. Ending FGM/C also contributes to achieving sustainable Development Goal 5.3 [23].

Despite the declining trends in FGM/C prevalence, the extent to which the CLARP model had a role to play in such declines in Kenya, and particularly in Kajiado county remains unclear. Consequently, to shed light on the contribution of CLARP in the fight against FGM/C, the present study sought to quantitatively and qualitatively evaluate the impact of the CLARP in Kajiado.

The main objectives of the evaluation were to: 1) Determine the impact of the community-led alternative rite of passage (CLARP) model on social and educational outcomes: Female Genital Mutilation/Cutting (FGM/C), child early and forced marriages (CEFM), teenage pregnancy (TP) and years of schooling among girls and 2) Explore the attitude, perception and practices of community stakeholders towards FGM/C.

## 2. Materials and methods

To quantify and document the impacts of the CLARP the study used a two-tier approach: a quasi-experimental approach which is quantitative, and a qualitative approach involving structured in-depth interviews (IDIs) focus group discussions (FGDs) and key informant interviews (KIIs) in addition to a detailed desk review of relevant documents.

## 2.1 Quantitative methods

**2.1.1 Econometric model.** The study employed a robust quantitative quasi-experimental method to quantify the impacts of the CLARP intervention. Specifically, the study used a difference-in-difference (DiD) approach to credibly quantify the impacts of the intervention which measures the average impacts of the CLARP roll-out on social and educational outcomes. The DiD approach calculates the average treatment effects by differencing the average outcomes before and after programme implementation across control and treated groups. The results were then compared with select counties with no CLARP before and after the intervention was rolled-out in 2009. FGM/C prevalence, educational outcomes, child early and forced marriages and teenage pregnancy data was abstracted from Kenya Demographic and Health Survey 2003, 2008–2009 and 2014.

The DiD approach estimates the average effects of CLARP programme on FGM/C prevalence and other social and educational outcomes before and after its rollout in 2009 across Kajiado versus two sets of control counties with high and low prevalence of FGM/C. The counties included were Mandera, Marsabit and Wajir which had the highest FGM/C prevalence as per 2003, 2008–2009 and 2014 KDHS data. A desk review of grey literature from the counties and consultation with existing non-governmental organizations indicated that such CLARP intervention or a similar one, as late as 2015, were non-existent in these counties. We also compared these to another panel of counties. These are Bungoma, Busia, Kakamega, Vihiga, Siaya, Homa Bay, Kisumu, Kilifi and Makueni which had both the lowest FGM/C prevalence and where the CLARP model was non-existent. These counties were test cases to explore any potential alternative explanations and run sensitivity checks to the baseline DiD estimates. To address potential concerns on the self-selection bias on the control counties, the analysis also explored three more potential control groups: (i) all counties with FGM/C prevalence rates less than 5 per cent, mostly counties in Western Kenya; and (ii) the baseline control counties plus Narok. As we show later in the key findings, the results are also robust to the first and second set of control counties.

Below is the DiD specification:

$$Y_{c,t} = \alpha_0 Post_t + \alpha_1 ARP_{c,t} + \alpha_2 Post_t \times ARP_{c,t} + X'_{c,t}\alpha_3 + District_d + Year_t + District_d \times Year_t + \epsilon_{c,t} \tag{1}$$

Where $c$ and $t$ stand for counties and years, respectively. $Y_{c,t}$ measures the various health and educational outcomes of interest (i.e., female genital mutilation/cutting, child early and forced marriages and teenage pregnancy as well as girls' years of schooling). The study exploited the rich KDHS individual records data which captures information on women aged 15–49 and extracted and analysed the following key survey variables: (i) whether the respondent is circumcised or not, (ii) whether the respondent was married below 18 years, (iii) whether the respondent gave birth below 20 years, and (iv) education in single years. Notice that (i) to (iii) capture latent binary variables while (iv) captures continuous variable. $Post_t$ is a binary indicator taking the value of 1 for years post 2009, and 0 otherwise. $ARP_{c,t}$ is also a binary indicator taking the value 1 if the CLARP was rolled out in a county in 2009, and 0 otherwise. $\alpha_0$ captures the separate average effects of time before and after the CLARP rollout. $\alpha_1$ captures the average effect of being in a county with the CLARP versus the county with none. $\alpha_2$ captures the average effect of the CLARP rollout. Variable $District_d$ stands for district/sub-county fixed effects which included unobserved district/sub-county characteristics (e.g., cultural beliefs and practices, the difference in districts/sub-counties responses' on FGM/C issues including the level of effort and resources deployed by sub-county authorities, and unobserved

geographical factors). $Year_t$ stands for year fixed effects which include unobserved year effects specific to the different counties. $District_d \times Year_t$ captures year specific district/sub-county unobserved effects. $\epsilon_{c,t}$ is an error term that captures all the other residual factors not captured in the specified model. To address potential intra-cluster correlations, the standard errors of all coefficient estimates are clustered at the county level. The vector $X_{c,t}$ captures time-varying observables that would otherwise confound the coefficient estimates of the CLARP roll-out effects. The control variables are: age at first cohabitation, age and sex of the household head, ethnicity, wealth index, age cohorts, residence (rural or urban) and religion. The inclusion of these variables attenuated the potential underlying bias that they could have on the average CLARP rollout effects.

**2.1.2 Quantitative data analysis.** To empirically investigate and respond to the first evaluation question, the study used KDHS data collected in 2003, 2008–2009, and 2014. The analysis used STATA 14 software. Calculations of FGM/C rates use KDHS binary indicator (0 = No, 1 = Yes) on responses of girls and women circumcision status. Education relies on school years in single digits while the construction of child early and forced marriages rates exploited the binary indicator (0 = Not Married, 1 Married at the time of being surveyed) and the age variable, allowing filtering of the sample to include the proportion of girls married below 18 years. Likewise, TP rates exploited binary indicator on current pregnancy status and age, thus generating proportions of young girls below 20 years who reported to be pregnant at the time of the survey.

Table 1 reports the summary statistics for girls and women aged 15–49 years across KDHS waves in 2003, 2008–2009, and 2014. The sample is divided into control (Wajir, Marsabit and Mandera counties) and treated (Kajiado county) groups, and the analysis compares the key summary statistics across the two groups. Except for FGM/C rates which are higher on control than in the treated groups, on average, the statistics indicate a close balance in both control and treated groups of the key outcome variables (schooling years, CEFM and TP) and their potential confounders.

## 2.2 Qualitative methodology

The study used qualitative methods to gain a deeper understanding of the quantitative results and on the status of CLARP in Kajiado County. Therefore, to answer the second and third evaluation questions, the study employed qualitative approaches to examine the socio-economic benefits that have accrued to CLARP beneficiaries and capture perceptions and attitudes of stakeholders in Kajiado County. As previously noted, the analysis and findings of the qualitative survey provide a better contextual understanding of the underlying facilitators and barriers to the success of the CLARP in Kajiado. These were carefully triangulated with the results from quantitative estimations.

**2.2.1 Sampling procedure and recruitment of study participants.** The study used a three-pronged approach to draw the sample for the qualitative methodology. For In-depths Interviews (IDIs), a random sample of CLARP beneficiaries (<5 years post-graduation) drawn within a school from a randomly drawn list of schools in both Magadi and Oloitoktok. The underlying assumption was that most CLARP beneficiaries in this category were students and still residents of both Magadi and Oloitoktok. CLARP beneficiaries with over five years post-graduation were randomly drawn from Amref's CLARP beneficiary's database and traced for interviews during the fieldwork. The stratification of respondents into recent (<5 years post-graduation) and long-time (>5 years post-graduation) graduates ensured that the interviews capture a broader range of experiences, especially from the latter category. Using Amref's database, the sampling filtered CLARP beneficiaries who joined and graduated from the programme between 2011 and 2016. A sample of 20 ARP graduates was drawn and stratified

**Table 1. Summary statistics, KDHS 2003, 2008–2009, and 2014.**

| Variables | Control group | | | Treated group | | | Overall | | |
|---|---|---|---|---|---|---|---|---|---|
| | Obs. | Mean | Std | Obs. | Mean | Std | Obs. | Mean | Std |
| Age | 2458 | 27.774 | 8.938 | 788 | 28.121 | 8.634 | 3246 | 27.858 | 8.865 |
| Age at first cohabitation | 1905 | 17.720 | 3.663 | 577 | 19.492 | 4.499 | 2482 | 18.132 | 3.944 |
| Age h/head | 2455 | 42.019 | 13.921 | 788 | 39.443 | 12.166 | 3243 | 41.393 | 13.559 |
| Sex h/head [1 = Female] | 2121 | 1.289 | 3.552 | 529 | 7.726 | 6.255 | 2650 | 2.574 | 4.952 |
| Circumcised [1 = Yes] | 1619 | 0.983 | 0.128 | 439 | 0.412 | 0.493 | 2058 | 0.862 | 0.345 |
| Education [years in school] | 2458 | 0.011 | 0.106 | 788 | 0.008 | 0.087 | 3246 | 0.010 | 0.102 |
| Married < 18 years [1 = Yes] | 2458 | 0.003 | 0.057 | 788 | 0.003 | 0.050 | 3246 | 0.003 | 0.055 |
| Pregnant < 20 years [1 = Yes] | 2458 | 0.012 | 0.108 | 788 | 0.010 | 0.100 | 3246 | 0.011 | 0.106 |
| Residence [1 = Rural] | 2458 | 0.424 | 0.494 | 788 | 0.320 | 0.467 | 3246 | 0.399 | 0.490 |
| Religion | 2445 | 1.98 | 0.034 | 787 | 0.0685 | 0.252 | 3232 | 1.786 | 0.03 |
| Ethnicity | 2445 | 5.963 | 0.026 | 788 | 1.981 | 0.003 | 3233 | 1.919 | 1.974 |
| Wealth Index | | | | | | | | | |
| Poorest | 2458 | 0.649 | 0.477 | 788 | 0.245 | 0.430 | 3246 | 0.551 | 0.497 |
| Poor | 2458 | 0.072 | 0.258 | 788 | 0.066 | 0.248 | 3246 | 0.070 | 0.256 |
| Middle | 2458 | 0.074 | 0.261 | 788 | 0.076 | 0.265 | 3246 | 0.074 | 0.262 |
| Rich | 2458 | 0.111 | 0.315 | 788 | 0.165 | 0.371 | 3246 | 0.124 | 0.330 |
| Richest | 2458 | 0.094 | 0.292 | 788 | 0.448 | 0.498 | 3246 | 0.180 | 0.384 |

Source: Authors' construction using 2003, 2008–2009, 2014 KDHS data

between recent (five years or less from graduation) and long-time (higher than five years) graduates of the CLARP programme ensuring that the interviews capture a broader range of experiences with the CLARP programme. To compare the differences in perspectives and experiences with the ARP model beneficiaries, additional four non-ARP beneficiaries' girls from the study communities were included totalling to 24 IDIs.

Sampling for Key Informant Interviews (KIIs) and Focus Group Discussions (FGDs) participants was homogeneous purposive sampling [24]. The respondents were selected purposively because of the different roles they play in the community that influence the FGMC practice and outcomes and homogeneous in terms of age (adolescent boys and girls) and gender among other variables. The respondents were identified through community engagement strategies coordinated by Amref team working in Kajiado. The process involved engaging local community leaders and stakeholders to inform them about the study and its objectives, deployment of local community mobilisers, as well as the implementation of community sensitisation *Barazas (public meetings)*. After introduction of the study, both KIIs and FGD respondents were identified by the consultants. The consultants identified 24 stakeholders to participate in KIIs and these included county government officials, local chiefs, religious and cultural leaders, children protection officers, community-based officers, members of county assembly, grandparents, TBAs, and teacher. A total of 14 FGDs (of 8–12 participants) were conducted with various sub-groups to ensure that the interactions within the sub-groups lead to a free and open expression of perspectives. FGDs participants included adolescent boys and girls, young men and women, *Morans*, and male and female parents. It was essential to include this group in the study as they influenced the delivery of the ARP model based on their gate-keeping role in the Maasai community. Traditionally, the *Morans* strongly believe that FGM/C is an essential cultural practice for keeping women and girls chaste. Ordinarily, a *Moran* will not marry a woman who has not undergone FGM/C. Besides being included in the sample,

eligibility to the study required potential respondents to be residents of the study areas and be willing to participate in the study voluntarily.

**2.2.2 Data collection.** The study used a theoretically grounded qualitative methodology to provide analysis of in-depth, existential accounts and perspectives of the sampled respondents using the standard qualitative data collection techniques: in-depth interviews (IDIs), KIIs and FGDs. IDIs applied to CLARP beneficiaries (women and girls aged 15 years and above who had graduated from the CLARP programme) and aimed at capturing the experiences and views of CLARP beneficiaries. As shown later in the results section, IDIs captured and documented individual stories, case narratives and respondents' status on outcome variables: FGM/C, child early and forced marriages, teenage pregnancies and educational outcomes.

KIIs and FGDs were employed to conduct detailed qualitative inquiries with various stakeholders to evaluate their experiences, perceptions, attitude and practices towards FGM/C. KIIs probed selected stakeholders to understand their interaction, working, success and suggestions for improving FGM/C interventions. FGDs probed cross-sections of key community members including the CLARP beneficiaries, young boys and girls, *Morans* and parents. All the qualitative data was collected in the month of September 2019. FGDs were conducted across individual sub-groups to ensure that the interactions within the sub-groups led to a free and open expression of perspectives [25]. Amref sub-contracted a team of independent consultants to facilitate collection and processing of qualitative data. Trained qualitative none Amref Health Africa research assistants and supervisors performed the IDIs, KIIs and FGDs in Kiswahili, *Ki-Maasai*, or English based on respondent's language preference. There was minimal contact of the Amref Health Africa staff with the study participants during the interviews hence minimizing the respondents bias. Semi-structured IDI, KII and FGD interview guides were used to collect the data. Interviews and discussions were audio recorded for transcription. The following data collection tools used are attached: IDI guide for ARP beneficiaries, and non-ARP Beneficiaries; KII guides with chief, member of County Assembly, Child protection officer, Headteacher; CBO Official, cultural elders, religious leaders, TBAs; FGD guides with Parents, *Morans*, young women, adolescent girls.

Except for a Member of County Assembly in Magadi and religious leader as well as young women FGD in Oloitoktok, the study surveyed all the targeted respondents totalling 62 interviews.

**2.2.3 Qualitative data analysis.** The researchers familiarised themselves with the data by reading the transcripts, from which they developed the coding scheme, later loaded in NVIVO 11 software for coding and analysis. The coding scheme built on prior themes from the literature review, research questions and interview guides. Additional nodes/codes that represented emerging themes not covered in the interview guides were added to the original codebook, constituting the grounded codes. A synthesis of the coded data using a framework matrix preceded thematic analysis to identify patterns of meaning or concepts that frequently occurred across interviews/coded data.

## 2.3 Study area

Kajiado County covers an approximate area of 21,000 sq.km and is adjacent to Nairobi, the capital city of Kenya. Kajiado county consists of several administrative districts including Kajiado Central, *Isinya*, *Oloitoktok*, *Magadi*, *Mashuru*, *Namanga* and *Ngong*. The sub-counties are Kajiado North, Kajiado West, Kajiado central, Kajiado East and Kajiado South. Magadi and Oloitokitok, where Amref rolled out the CLARP for the first time in 2009, were used for qualitative surveys.

The county has an estimated population of about 1,117,840 people as per 2019 census, a total of 316,179 households and an average household size of 3.5 [26]. The economy thrives

mainly on agriculture and related activities including food crop farming, livestock production, dairy and beef production, hides and skins, poultry and horticulture.

### 2.4 Ethics approval and consent to participate

Ethics approval for this study was acquired from Amref Ethical and Scientific Review Committee, with reference number:—AMREF-ESRC P677/2019. All individual interviews were conducted in a private setting. The fieldwork team provided the participants with an Informed Consent Form (ICF) in Kiswahili/*Ki-Maasai*/English (depending on the language the participant is most comfortable with). The ICF included a section for seeking the assent of participants under the age of 18 years but above 15 years in addition to seeking consent from parents to the minors. Interviews were only conducted for participants who signed the ICF. Interviews were only recorded after obtaining permission from participants.

### 2.5 Study limitations

The quantitative analysis relied on slightly dated data between 2003 and 2014 because current data for 2018 is unavailable. The estimated impacts of CLARP may have worked during the period 2003–2014 only. While the quantitative analysis resonates with the attribution aspect, the analysis focuses on the entire Kajiado County with the assumption that Amref rolled out CLARP in the whole of Kajiado County, though this was not the case. Therefore, the analysis provides upper-bound estimates as it encompasses the entire County while intervention rollout was done in specific hotspots of Magadi and Oloitokitok. Moreover, the direct attribution to the Maasai community is hard to make because of the presence of multiple ethnicities that live in Kajiado and practice FGM/C. Additionally, we could not use the parallel trends assumption test due to data gaps. We instead did a balance check of the outcome variables before the introduction of the intervention. The aspect of reporting on practices using qualitative interviews is limiting as the practice is done in secrecy. This was however mitigated through assuring and re-assuring study participants during consenting that any information shared will fully be kept confidential and that the study team will not discuss any information about them or what they shared unless with written permission. Lastly, the DiD approach used for the quantitative analysis may not be the best method to tease out the impacts of the CLARP intervention given that it is a quasi-experimental design. It is however, the most appropriate design given that experimental design was not integrated in the programme right from the start of the implementation. This would have been impractical to do so given that we did not have control over implementation of similar activities by other organizations.

## 3. Results

We present both the quantitative and qualitative findings regarding the impacts of CLARP on FGM/C prevalence, CEFM, TP and years of schooling among girls in Kajiado County. The Pre-CLARP intervention balance check of outcome variables is presented in Table 2 where the test of means is based on estimated residuals of regression equations with all four outcome variables.

The result indicates no statistically discernible differences in the levels of outcome variables before the introduction of the CLARP in Kajiado.

Table 3 is a summary of difference-in-difference regression results.

Column 1 shows the estimated outcomes of interest. Column 2 provides averages of the outcome variables between the control group counties (i.e., Mandera, Marsabit, and Wajir) and Kajiado (the treated County) before the rollout of the CLARP. Columns 3 and 4 report the estimated average treatment effects following, consecutively, the magnitude of the average

**Table 2. Pre-ARP intervention balance check of outcome variables.**

| Variables | Control group | Treated group | Mean Difference |
|---|---|---|---|
| FGM/C rates | -0.00151 | -0.000123 | -0.00138 |
| Schooling years | -0.0260 | -0.00556 | -0.0204 |
| CEFM rates | 0.000708 | -4.00e-06 | 0.000712 |
| TP rates | -0.000305 | -0.000130 | -0.000175 |

effect and the sign of the effect. Columns 5 report the percentage impact change–calculated as a mean of the estimated average effect divided by the overall mean pre-CLARP rollout–(reported as a factor change) on all outcome variables. Overall, the reported results indicate the average treatment effects before and after CLARP rollout in Kajiado compared to control counties. See Table 4 for all Difference in Difference regression results.

The summarised results in Table 3 were estimated from fully specified models controlling for all relevant variables that can, in theory, bias the estimated average treatment effects. Theory and existing literature guided the inclusion of these variables in the analysis [1, 9, 16, 27–29]. The potential variables are the age of respondents, age and gender of household head, age at first cohabitation, residence (rural or urban) status, religion, ethnicity and wealth index included in the analysis. For coherence and clarity, the results are presented in four themes as below.

## 3.1 Impact of CLARP on FGM/C

The results suggest that FGM/C prevalence declined by 24.2 per cent in Kajiado county compared to Wajir, Mandera, and Marsabit (p<0.10). The analysis of qualitative findings also corroborates the reported quantitative impacts on FGM/C. Qualitative findings indicate declining FGM/C prevalence in both Magadi and Oloitokitok. Study participants generally regarded CLARP as one of the significant contributors to the declining levels of FGM/C in the study communities.

> *"We would like to tell Amref to continue because we have realized we have moved a step for-ward. Concerning FGM/C, we have moved a step because it has reduced though hasn't stopped. It is no longer at the same level as it was previously because it was very high. It was not a secret but the ones currently practicing it are doing it at night as we had previously dis-cussed and taken to other locations for cutting. Therefore, we are requesting Amref to continue with their work because if they do, FGM/C will stop eventually".* ~**FGD** *Morans.*

The findings show that there was a sense that Amref need to continue with the intervention for FGM/C to continue to decline. The CLARP also appears to have generally contributed to a paradigm shift in the community by normalising the open conversation about the culture of FGM/C, increasing enlightenment about its harms, changing mind-sets and empowering fam-ilies to make better decisions.

**Table 3. Summary of difference-in-difference regression results.**

| Variables | Pre-CLARP rollout | Estimated Treatment Effects | | Impact change |
|---|---|---|---|---|
| | Mean | Mean | Sign | (factor) |
| Rates of Female Genital Mutilation/Cutting | 0.808 | 0.242 | -ve | 0.30 |
| Education in single years | 3.124 | 2.469 | +ve | 0.79 |
| Rates of child early and forced marriages | 0.012 | 0.049 | -ve | 4.08 |
| Rates of child early teenage pregnancies | 0.015 | 0.063 | -ve | 4.20 |

**Table 4. Difference in difference regression results.**

| Dependent variables | Baseline | | | | Alternative control counties #1 | | | | Alternative control counties #2 | | | |
|---|---|---|---|---|---|---|---|---|---|---|---|---|
| | (1) | (2) | (3) | (4) | (1) | (2) | (3) | (4) | (1) | (2) | (3) | (4) |
| | FGM/C | Education | Marriage | Pregnancy | FGM/C | Education | Marriage | Pregnancy | FGM/C | Education | Marriage | Pregnancy |
| Post x Treat | -0.242* | 2.469** | -0.049*** | -0.063*** | -0.133* | 2.479** | -0.157*** | -0.195*** | -0.195 | 4.318*** | -0.045*** | -0.055*** |
| | [0.133] | [1.218] | [0.015] | [0.021] | [0.078] | [0.964] | [0.022] | [0.016] | [0.152] | [1.122] | [0.016] | [0.019] |
| Education [years] | -0.011*** | | -0.001 | -0.000 | -0.011*** | | -0.001 | -0.002 | -0.013* | | -0.001 | -0.000 |
| | [0.004] | | [0.001] | [0.003] | [0.004] | | [0.001] | [0.002] | [0.007] | | [0.001] | [0.002] |
| Age | -0.015** | -0.076 | -0.039*** | -0.038*** | -0.026*** | -0.188* | -0.039*** | -0.036*** | -0.016* | -0.037 | -0.035*** | -0.036*** |
| | [0.006] | [0.075] | [0.010] | [0.008] | [0.009] | [0.108] | [0.011] | [0.008] | [0.009] | [0.080] | [0.009] | [0.007] |
| Age^2 | 0.000** | 0.001 | 0.001*** | 0.001*** | 0.000*** | 0.002 | 0.001*** | 0.000*** | 0.000*** | -0.000 | 0.000*** | 0.001*** |
| | [0.000] | [0.001] | [0.000] | [0.000] | [0.000] | [0.002] | [0.000] | [0.000] | [0.000] | [0.001] | [0.000] | [0.000] |
| Gender h/head [1 = Female] | 0.020 | -0.362 | -0.000 | -0.009 | 0.002 | 0.019 | 0.012 | -0.002 | 0.017 | -0.158 | 0.002 | -0.011 |
| | [0.013] | [0.251] | [0.007] | [0.009] | [0.015] | [0.209] | [0.008] | [0.007] | [0.021] | [0.251] | [0.006] | [0.008] |
| Age h/head | 0.003 | -0.001 | 0.006** | 0.002 | 0.005 | 0.006 | 0.007* | 0.003 | 0.009* | 0.001 | 0.006** | 0.002 |
| | [0.002] | [0.052] | [0.003] | [0.003] | [0.004] | [0.042] | [0.004] | [0.002] | [0.005] | [0.058] | [0.003] | [0.002] |
| [Age h/head]^2 | -0.000 | 0.000 | -0.000** | -0.000 | -0.000 | -0.000 | -0.000 | -0.000 | -0.000* | 0.000 | -0.000** | -0.000 |
| | [0.000] | [0.000] | [0.000] | [0.000] | [0.000] | [0.000] | [0.000] | [0.000] | [0.000] | [0.001] | [0.000] | [0.000] |
| Age at marriage | -0.022 | 0.054 | -0.007** | -0.001 | -0.032*** | 0.630*** | -0.004 | 0.001 | 0.005 | 0.341** | -0.005* | -0.000 |
| | [0.015] | [0.126] | [0.003] | [0.004] | [0.011] | [0.148] | [0.003] | [0.002] | [0.031] | [0.133] | [0.003] | [0.004] |
| [Age at marriage]^2 | 0.001 | 0.002 | 0.000** | 0.000 | 0.001*** | -0.009** | 0.000 | -0.000 | -0.000 | -0.004 | 0.000* | -0.000 |
| | [0.000] | [0.003] | [0.000] | [0.000] | [0.000] | [0.004] | [0.000] | [0.000] | [0.001] | [0.003] | [0.000] | [0.000] |
| Residence [1 = rural] | 0.054 | 1.568** | 0.028*** | 0.028** | -0.015 | 1.712** | 0.018 | -0.006 | 0.123** | 1.490* | 0.022** | 0.019** |
| | [0.048] | [0.743] | [0.009] | [0.011] | [0.042] | [0.697] | [0.016] | [0.013] | [0.052] | [0.770] | [0.009] | [0.009] |
| Constant | 0.724*** | 8.491*** | 0.695*** | 0.809*** | 0.739*** | -2.939 | 0.516*** | 0.442*** | 0.492 | 3.367 | 0.624*** | 0.783*** |
| | [0.231] | [2.554] | [0.178] | [0.158] | [0.205] | [2.620] | [0.122] | [0.102] | [0.356] | [2.886] | [0.157] | [0.142] |
| Religion | Yes | Yes | Yes | Yes | Yes | Yes | Yes | Yes | Yes | Yes | Yes | Yes |
| Ethnicity | Yes | Yes | Yes | Yes | Yes | Yes | Yes | Yes | Yes | Yes | Yes | Yes |
| Wealth Index | Yes | Yes | Yes | Yes | Yes | Yes | Yes | Yes | Yes | Yes | Yes | Yes |
| Year FE | Yes | Yes | Yes | Yes | Yes | Yes | Yes | Yes | Yes | Yes | Yes | Yes |
| District/Sub-county FE | Yes | Yes | Yes | Yes | Yes | Yes | Yes | Yes | Yes | Yes | Yes | Yes |
| District/Sub-county x Year FE | Yes | Yes | Yes | Yes | Yes | Yes | Yes | Yes | Yes | Yes | Yes | Yes |
| Observations | 1120 | 1749 | 1749 | 1749 | 795 | 1382 | 1382 | 1382 | 1219 | 1916 | 1916 | 1916 |
| R-squared | 0.895 | 0.850 | 0.271 | 0.181 | 0.739 | 0.757 | 0.291 | 0.214 | 0.795 | 0.814 | 0.265 | 0.188 |
| Average Dep. Var | 0.808 | 3.124 | 0.012 | 0.015 | 0.166 | 6.085 | 0.013 | 0.013 | 0.782 | 3.842 | 0.011 | 0.014 |
| S.d. Dep. Var | 0.394 | 5.505 | 0.110 | 0.122 | 0.373 | 5.671 | 0.112 | 0.113 | 0.413 | 5.589 | 0.103 | 0.119 |
| Percentage impact change (factor) | -0.30 | 0.79 | -4.08 | -4.20 | -0.80 | 0.41 | -12.08 | -15.00 | -0.25 | 1.12 | -4.09 | -3.93 |
| Co-efficient estimates | -0.242 | 2.469 | -0.049 | -0.063 | -0.133 | 2.479 | -0.157 | -0.195 | -0.195 | 4.318 | -0.045 | -0.055 |
| Average outcome variable (control group) | 0.808 | 3.124 | 0.012 | 0.015 | 0.166 | 6.085 | 0.013 | 0.013 | 0.782 | 3.842 | 0.011 | 0.014 |

*(Continued)*

**Table 4.** (Continued)

| Dependent variables | Baseline | | | | Alternative control counties #1 | | | | Alternative control counties #2 | | | |
|---|---|---|---|---|---|---|---|---|---|---|---|---|
| | (1) | (2) | (3) | (4) | (1) | (2) | (3) | (4) | (1) | (2) | (3) | (4) |
| | FGM/C | Education | Marriage | Pregnancy | FGM/C | Education | Marriage | Pregnancy | FGM/C | Education | Marriage | Pregnancy |
| %age impact change | -0.30 | 0.79 | -4.08 | -4.20 | -0.80 | 0.41 | -12.08 | -15.00 | -0.25 | 1.12 | -4.09 | -3.93 |

Standard errors are in brackets and clustered at the regional level.

"* p<0.10

** p<0.05

*** p<0.01"

Baseline control counties: Wajir, Mandera and Marsabit

Alternative control counties #1: Control counties are all counties with low prevalence <5%

Alternative control counties #2: Control countries include Narok, Wajir, Mandera and Marsabit

*"Today, they find it normal. However, it was difficult for leaders to speak against the culture. The community always thought that the person was introducing western civilization and were always up in arms. Things are now working as many people are now ready to hear and even attend the training. After the training, the people can now make better decisions."* **KII Member County Assembly**

*"It has been a good program. Making of decisions to stand up against this culture which has been destroying many lives was something that many people could not make on their own. The training has made it possible. We are now leaving the culture that is not fruitful to us and moving on well. People have realized that the old culture is not of benefit."* ~**IDI long-term CLARP beneficiary**

### 3.2 Impact of CLARP on child early and forced marriage (CEFM) rates

Quantitative analysis shows that CLARP had discernible impacts on CEFM rates in Kajiado compared to control counties. The estimates show that CEFM declined by 4.9 per cent ($p < 0.01$).

The analysis of CEFM using the qualitative data shows that CLARP models helped girls and women to make more informed decisions on delaying marriage and childbirth. The analysis shows that all the interviewed CLARP beneficiaries indicated that they would only think about marriage once they have completed their education. Usually, when a girl experiences FGM/C, they are now considered by the community to be mature and ready for marriage. However, CLARP appears to have empowered its beneficiaries to choose when to get married.

*"For me not being circumcised, I will choose on the time to be married. I will continue with my studies up to where I will be able to reach that will make me choose the best person that I have liked to marry me. But for the person that is cut, immediately after cut, your father know that you are now mature to be married. He can give you out without completing your studies and even choosing the person that you don't liked."* ~**IDI long-term CLARP beneficiary.**

*"Being uncircumcised will help me plan for my future. It will help me finish my schooling on time and get a job before I choose who to marry me at the right time and start having children. But the circumcised she will be married off at the age of 15 or 16 to a man not of her choice and start having children as early as possible."* ~**IDI long-term CLARP beneficiary.**

### 3.3 Impact of CLARP on teenage pregnancy (TP) rates

CLARP also appear to have had measurable quantitative impacts on TP rates in Kajiado compared to control counties. The analysis reveals that TP declined by 6.3 per cent ($p < 0.01$). Qualitative analysis also backs up this finding. Majority of the respondents indicated that the CLARP intervention had led to a decrease in the number of teenage pregnancies.

*"From my observation, it has reduced but still exist. Previously, teenage pregnancy was three or two every term but has dropped to two or three in a year."* ~**FGD respondent, male parents.**

This apparent reduction in teenage pregnancy was linked to the reduced likelihood of CLARP beneficiaries to be married off early as compared with non-beneficiaries. CLARP beneficiaries were also said to have higher self-worth and consequently, a lower likelihood of engaging in risky sexual behaviours.

*"I will say that due to the awareness especially the CLARP programs, the trainings, pregnancy has really gone down especially when the trainings that are conducted, they really encourage a lot of girls and make someone among themselves and the issue of self-worth that they take pride among themselves when they don't have those sexual relationships with boys." ~***KII Children Protection Officer**.

However, not everyone agreed that CLARP had any differential impact on TP. One participant mentioned that although the CLARP model has led to a reduction in teenage pregnancies, there are still some uncut girls that become pregnant because of peer pressure.

*"We still experience teenage pregnancies. That is something that the nurses and CLARP teachers needs to include now, to say even if you are a woman through CLARP, it doesn't guarantee you to become a woman and give birth. But for now, I can say it is the same." ~***KII Community-based Organisation Officer.**

## 3.4 Impact of CLARP on educational outcomes

Quantitative analysis show that CLARP had a tangible impact on improving educational outcomes in Kajiado compared to control counties. The estimates suggest that CLARP interventions in Kajiado resulted in an average increase in schooling years for girls and women by roughly 2.5 years (p<0.05).

The qualitative results show that there was a consensus that CLARP beneficiaries were more likely to complete school as compared to the non-beneficiaries.

*"What I can say is, since I joined this school in 2010, school completion has improved because I think that is the same year CLARP started. They complete school and join secondary schools. This type of education has brought a lot of change". ~***FGD respondent, male parents.**

*"My father was suggesting circumcising me and marry me off the same thing he did to my older sisters, but when I attended the training, he tried to marry me off, but I refused. When he realized that I was not changing my mind he started supporting me with my education and he told me, learn until you reach where you want to reach." ~***IDI long-term CLARP beneficiary.**

The increased likelihood of school completion among CLARP beneficiaries was attributed primarily to the decreased likelihood of early marriage. CLARP beneficiaries appear to have a stronger agency to resist early marriage so that they could complete school.

*". . .There is difference. Because the circumcised will be married between 15 years, 16 years, or, 17 years but am in school, until I complete my school, that's when I will think about marriage." ~***IDI long-term CLARP beneficiary.**

## 3.5 Perceptions, attitudes and practices of community stakeholders about FGM/C practices

This section reports the analysis on the perceptions, attitudes and practices of community stakeholders about FGM/C practices in Kajiado. The findings show a strong consensus that the study communities embraced the CLARP model because of its demonstrable impacts in the lives of its beneficiaries and their families. However, there remain some barriers to the model's full effectiveness. These barriers correlate with the perceptions, attitudes and practices of community stakeholders about FGM/C. This section highlights some of these barriers. It

also identifies opportunities to refresh and reinvigorate the CLARP for better and more sustainable impact.

**3.5.1. Barriers.** *a) Resistance to cultural change.* Most of the study participants said they do not see any risk or disadvantage of the CLARP model. However, a few people mentioned that the key barrier to the program is the fact that it is still perceived to go against the *Maasai* culture.

> *"According to the community's belief, they still believe it is important. However, much you try to convince them that it is outdated, they cannot give up the practice. Many, especially the elderly will ignore you".* ~**IDI long-term CLARP beneficiary.**

The *Morans*, young men and elderly community members (such as grandparents) were particularly resistant to CLARP because they believed in the sacredness of the FGM/C practice even though they tended to accept that the practice did not provide any tangible benefits.

> *"This is because of our culture, what I can say is that even the young men are propagating the spread of the FGM/C practice in this community. Most of us do not want to marry uncut girls. For instance, if I got married and during lovemaking, I get to realize that the woman I got married to be not circumcised, I will send her back to her parents".* ~**FGD adolescent boys.**

*b) Persistent stigma.* One daunting consequence of the resistance to cultural change is the persistence of stigma. Girls who do not undergo the cut, including those that graduate from the CLARP training remain at high risk of ridicule, discrimination and rejection in the community. Uncircumcised girls/women are looked down upon by the community and endure name-calling using such derogatory terms as *'entaapai'* (uncircumcised). Regardless of their age or educational status, uncircumcised girls are considered by many to be children since they have not undergone the traditional rite of passage into womanhood. In some instances, society looks at them as *non-Maasai* or aliens.

> *". . .Some of us we are big girls but are going to school. We seat with circumcised girls in class. They under look at us only that they consider themselves as women and we are a girl. They will not dare even seat with you".* ~**IDI recent CLARP beneficiary.**

> *"Concerning this CLARP, we have some Morans who are against it because they believe if you are not circumcised it won't be good at all; because if you marry uncircumcised woman you will be staying with a girl in the house, and that is not good".* ~**FGD adolescent boys.**

*c) Peer-pressure.* CLARP beneficiaries face significant peer-pressure, particularly from young men as well as girls that have undergone the cut. At the core of this peer-pressure are the issues of dating and marriageability. Generally, male suitors prefer girls that have undergone the cut. These girls waste no time in flaunting the attention and gifts they receive.

> *". . .In addition to that, there is no man who would marry anyone who has not undergone FGM/C. When they listen to such advices, they go to class telling their classmates how they feel proud holding each other's hands. Those who have undergone the FGM/C only sit with those who have undergone through FGM/C. If they have boyfriends, they sit and walk together holding hands. Those who have boyfriends have a lot of advantages. This makes those without boyfriends to go for the FGM/C in order to have the advantages."* ~**IDI, CLARP recent beneficiary.**

*"When we meet, some girls are circumcised. They will inform you that you can't get a boy-friend unless you are circumcised. They will advise you to get circumcised first. Sometimes you can be convinced by seeing gifts they get from their boyfriends. That might lead you to get circumcised so that you can also get a boyfriend who will give you gifts. That is how it is."* ~**IDI, CLARP recent beneficiary.**

*"Yes, girls that are cut and those who are not cut they say their dowry is not the same. A girl who is cut has already passed the rite of passage so her dowry will be high since she has not engaged with other men. They say that the girl maybe a virgin".* ~**IDI long-term ARP beneficiary.**

*"The parent would want to get money quickly because he will not if the daughter is not cir-cumcised."* ~**FGD parents.**

*d) A rise in secret circumcisions.* One worrying trend that was highlighted by several study participants was the rising practice of FGM/C in secret. The illegality of FGM/C was driving several covert practices including conducting the cut in the dead of the night or sending the girls across the border into Tanzania.

*"Some candidates are taken as far as out of this country like in Tanzania, or at the borders there they are circumcised secretly, and they stay there till when they are healed. Apart from that, they may go to reside with some close relatives say grandparents where this act may be conducted. These arrangements are made by the parents who want their children to be cir-cumcised but they are afraid of the authorities."* ~**CLARP long-term beneficiary.**

*"The traditional circumciser will be called to circumcise a girl at home especially at night and the girl is forced to stay indoors until she is fully healed, in case a neighbour visits the home-stead he/she is not be aware that there is a girl who has undergone FGM/C."* ~**FGD Adoles-cent boys**.

**3.5.2. Opportunities to improve/adapt.** *a) Correcting misperceptions about uncircumcised girls*. There remain several misperceptions about uncircumcised girls, especially among young men and *Morans*. For example, uncircumcised girls were reported to be more sexually active and tend to conceive faster than those who have undergone the cut. Some respondents highlighted that uncircumcised girls were promiscuous, and hence men prefer the circum-cised, less promiscuous women. Education and awareness programmes that target the misin-formed groups can correct such misconceptions.

*"It is the truth because the uncircumcised girl is very loose because her blood is still hot. If you try to seduce them, you will notice the difference between a circumcised and uncircumcised girl. This is because the circumcised girls are very difficult to seduce but for the uncircumcised it is different."* ~**FGD Morans.**

*"You know the circumcised one lost a lot of blood during circumcision therefore the uncircum-cised has more heat compared to the other, therefore can conceive quickly."* ~**FGD Morans.**

*b) Highlighting economic empowerment.* Some study participants highlighted that circumci-sion is associated with economic gains because girls are married off soon after the circumci-sion. Thus, the dowry payment was highlighted as an economic motivation for parents, especially fathers to have their daughters circumcised in readiness for marriage and in

anticipation for dowry payment. It also emerged that the dowry price for circumcised girls is higher than that of uncircumcised girls.

> "...CLARP programme also has helped them because they have known the importance of not circumcising their girls. As for now we have girls who go to school, some have jobs and help their parents. Before girls were being circumcised and married off then dowry is paid. Latter those cattle die due to drought. But now they are bringing salary. Whatever small they get they share with their parents. Now parents have seen the importance of not circumcising their girl through the CLARP model." ~**FGD adolescent boys.**

However, the narrative appears to be changing. The community is now witnessing a new economic phenomenon involving uncircumcised girls that have completed their studies and gone on to pursue successful careers. These girls then support their parents financially. Given the feedback that the CLARP intervention improved school completion rates among its beneficiaries, therein lies an opportunity for the model to educate the community about the long-term economic empowerment of girls that have passed through the programme.

*c) More visibility to the harms of FGM/C.* Another opportunity lies in further educating the community members about visible harms of FGM/C. Young boys appeared quite knowledgeable about these harms, which they mentioned to include high risks of sexually transmissible diseases, complications during childbirth and school drop-out. These young men could be current and future ambassadors of the CLARP interventions to the broader community.

> "FGM/C negatively impacts the lives of the girls because during FGM/C they destroy part of woman's body which in turn brings a problem when these girls are giving birth." ~**FDG adolescent boys.**

> "As a Maasai community, we have experienced a lot of challenges in relation to FGM/C because once a girl is circumcised, she drops out of school because she feels she is mature enough to be married." ~**FDG adolescent boys.**

Other ambassadors of change could include the TBAs who were typically the people responsible for executing the cut. The analysis suggested a declining role of TBAs and that some of them are embracing a new way of thinking.

> "I refused and instead joined the church. I told the community I will not do that again" ~**KII TBA**.

> "It has made them respect me more...They like me because I have stopped what they also don't like." ~**KII TBA**.

*d) Follow-up and reaching excluded groups.* According to some study participants, improvements of the CLARP model would require reaching out to specific groups that were not adequately involved in the implementation of the intervention. These include small community-based groups, older men and women, parents and men who are the household heads. They also stressed the need for follow-up with former CLARP participants to ensure they adhere and live by CLARP training expectations consistently to help beneficiaries not to succumb to peer pressure and backslide to becoming FGM/C victims.

> "What I am requesting you is, try and set aside a week to train the old men because that is the only time, they will understand what Amref is talking about. Let us not take them for a joke.

*If we approach them as young men, they will not take us seriously but if you people approach them and they see us together they will take the issue seriously". ~**FGD *Morans.***

*"This is due to peer pressure because if a girl is not circumcised and she comes back to a community where majority of the girls have been circumcised, she will be ridiculed by her peers hence the pressure to undergo FGM/C." ~**FGD Adolescent girls.**

*e) Law enforcement and political will.* It was evident from the interviews that the study communities were aware of the laws of the land in so far as FGM/C is concerned. Several participants mentioned that the risk of arrest and prosecution had served as a significant deterrent to FGM/C in their communities. However, some participants were unconvinced that the laws were being enforced adequately, especially by some local chiefs. Others were worried about being stigmatised for cooperating with law enforcement officials. Future CLARP interventions should, thus, consider how best to work with government officials to socialise and implement the respective anti-FGM/C laws.

*"FGM/C stopped though our wives are circumcised. It is not practiced anymore in this area because of the training that have been conducted and the government is also against it. If any parent now practices FGM/C he or she will be arrested and taken to jail." ~**FGD *Morans.***

*"I know it is unconstitutional and unlawful and you can be charged in the court of law. But in the real sense we have never seen or heard a person who has facilitated the proceeding taken to court of law. I can say they are there but silent." ~**FGD Young men.**

## 4. Discussion

Existing empirical evidence on the interplay between FGM/C and ARP models indicate that the latter plays a significant role in shifting knowledge, attitudes and perceptions of FGM/C practices in a society [1, 6, 27, 29–32]. By catalysing influence on its abandonment, ARP model have directly and indirectly contributed to the declining FGM/C's prevalence rates, improved educational outcomes for girls, and reduction in child early and forced marriages and teenage pregnancies [1, 2, 6, 33, 34] assert that the effectiveness of the ARP models hinges on its community-led approach in changing social norms on FGM/C, as well as empowerment of girls and women through education.

Much of the existing literature on the evaluation of ARP models have centred on their impacts on community knowledge, attitudes and perceptions [1, 6, 27, 29–32]. Nevertheless, there is growing recognition that ARP can have farfetched impacts beyond changing the social norms and behaviours on FGM/C [3, 14, 31]. Despite these recognitions and widely celebrated impacts of ARP models in changing societal norms and behaviours towards FGM/C, scant evidence exists on the quantitative and qualitative impacts of ARP models in not only reducing FGM/C prevalence but also in reducing child early and forced marriages, teenage pregnancies and improvement in educational outcomes. The present evaluation study bridges the existing knowledge gap by quantitatively and qualitatively evaluating the impacts of the CLARP model in Kajiado county in Kenya.

Employing quasi-experimental and qualitative methods to sleuth and analyse detailed quantitative qualitative survey data, the findings of this study lend credence to the claim that the rollout of the CLARP model reduced FGM/C practices, child early and forced marriages, teenage pregnancies and improved girls' schooling years in Kajiado. Quantitatively, the estimated magnitude of impacts indicates that CLARP reduced FGM/C by 24 per cent, reduced early and forced child marriages and early teenage pregnancies four-folds over,

respectively, and increased schooling years by 2.5 years. Further analysis of KDHS data also broadly corroborate these findings: not only have FGM/C prevalence been on the decline as depicted in Fig 1, but CEFM and TP have also been on the decline. CEFM declined from an average of 1.22 per cent in 2003 to 0.37 per cent in 2014. TP declined from an average of 1.29 per cent in 2003 to 0.38 per cent in 2014. On the other hand, the average years of schooling increased from 6.87 years in 2003 to 9.3 years in 2014.

The findings of this study are in line with other studies [35, 36] which point to barriers to the adoption of alternative models to FGM/C such as the CLARP. *Morans*, young men and the elderly in the community showed resistance to CLARP which lead to persistent discrimination and stigmatization of girls who do not undergo the cut. The finding that dowry price for circumcised girls is perceived to be higher than that of uncircumcised girls is also reflected in a similar study in Nigeria [37] which establishes that FGM raises the social status for the family and generate income when the daughter gets married and the dowry is paid.

Despite the reported barriers, qualitatively CLARP interventions appear to not only have left positive experiences to its beneficiaries but also improved their lives. The analysis also shows that the CLARP has several perceived benefits to its beneficiaries: empowering girls to make decisions to delay marriage and childbirth and thus pursue the accumulation and development of their human capital. Finally, the findings reveal that *Maasai* communities in two sub-counties of Kajiado (Oloitokitok and Magadi) have embraced the CLARP model because of its tangible positive effects on young girls' lives in these communities.

The findings reported in this study are also consistent with those of the *Yes I Do Alliance* (YIDA) programme [38] which also point to a reduction in child marriages, teenage pregnancies and FGM/C prevalence as a result of community engagement and sensitisation on gender equality and sexual and reproductive health knowledge. YIDA's interventions aim at promoting gender-transformative thinking, girls' empowerment and engagement of men and boys as gatekeepers. The five (5) year programme (2016–2020) worked with vulnerable communities to end FGM/C, teenage pregnancies and child marriages in Ethiopia, Kenya, Malawi, Mozambique, Zambia, Pakistan and Indonesia. In Kenya, the programme worked within the *Maasai* community to provide an ARP to girls, with more than 1,300 girls having been trained and graduated from the programme. A mid-term review of the programme indicated a positive influence on community perspectives on deeply rooted cultural norms such as FGM/C, with community members appreciating the importance of girl child empowerment through education, which consequently is delaying child marriages and teenage pregnancies.

The findings of the present study have important policy implications for policymakers, health development practitioners and Amref Health Africa. The findings provide persuasive evidence that CLARP model not only works but has discernible impacts on its beneficiaries in Kenya providing scope for scaling-up the models to other areas and countries where FGM/C practices are widespread and rampant. By extension and in line with other studies [1, 32, 33, 39, 40] the results suggest that harnessing and mainstreaming CLARP interventions into, for example, education curricular and community-led outreach programmes, can have more tangible and sustainable impacts on both its beneficiaries and the society at large.

Efforts by governments, multi-lateral organisations and community-led initiatives have driven recent declines in FGM/C rates in Africa [18, 41]. The fact that CLARP interventions have had positive impacts in Kajiado gives hope, at least on the margin, for yielding substantial results if scaled-up. Presumably, the positive CLARP effects signal the need for, amongst others, more resource mobilisations to entrench the interventions further interior to the rural and grassroots areas to effectively fight the persistent FGM/C practices standard in such places.

Typically, FGM/C, child early and forced marriages, teenage pregnancies and girls' educational outcomes are interlinked [6, 9, 14, 28, 42]. Thus, the documented positive findings

suggest that the CLARP model can act as a replacement for FGM/C rituals as a rite of passage and at the same time be an educational and empowerment conduit for empowering girls and women in Kenya. The findings also signal the potent role CLARP have in lessening and curtailing the alarming rates of child early and forced marriages [16, 42] and thus contain its spill over effect to teenage pregnancies [14] minimising maternal death and childbirth complications associated with teenage pregnancies [9, 43]. Spearheading the FGM/C fight in Kajiado, alongside governments, local and international, Amref Health Africa's CLARP model, is evidently, a force to reckon. The contribution of CLARP interventions in the fight against FGM/C practices and their related manifestations suggest more resources and collaborative efforts from other stakeholders are needed to scale-up the CLARP interventions to eradicate FGM/C.

Combining quantitative and qualitative approaches to impact evaluation contributes to moving methodological discussions away from a norm in development research in which qualitative research plays 'second fiddle' to conventional empiricist investigation [44]. We have very well combined the two methodologies in this study where we have used qualitative findings to observe, corroborate and justify impacts based on local context and relevance for policy audience. Engaging a team of independent consultants to lead the data collection and processing of the data in addition to thoroughly training the team of research assistants on data collection processes. This reduced contact between Amref staff and the study participants during data collection hence minimizing respondents bias. In addition, engaging a team of independent consultants promoted objectivity and reduced interpretation bias of the findings.

## 5. Conclusions

The study revealed that CLARP played a significant role in attenuating FGM/C, CEFM and TP in Kajiado County. The results also indicate that the intervention increased schooling years for girls in the county. Furthermore, the findings show that the CLARP is slowly being embraced by the *Maasai* communities in Magadi and Oloitokitok sub-counties and its implementation has left positive experiences on the ground. Additionally, the intervention empowered girls and women to claim and protect their social and human rights and accumulate more human capital, aside from further improving their lives. It will be important to re-run the analyses when the 2018 KDHS data will be available to see whether observed impact still hold. CLARP is replicable as it is currently being implemented in Tanzania. We recommend scaling it up for adoption by stakeholders implementing in other counties that practice FGM/C as a rite of passage in Kenya and across other sub Saharan Africa countries.

## Supporting information

**S1 Data.**
(DOCX)

## Acknowledgments

We thank all the research assistants who worked tirelessly to ensure that the data collected was of quality. These are Lester Linti, Joshua Kelele, Gideon Kanchori, Felistus Nashipae, Rachel Matayen, Alex Sananka, Thomas Tuke, Janet Seleina and Rachel Lemaikai. We are also grateful to Amref Health Africa drivers:- Samuel Mukabana and Rufas Njuguna for ably driving the field team through the rough terrains of Kajiado County.

**Disclaimer:** The views, thoughts, and opinions expressed in this paper belong solely to the authors, and not necessarily to the author's employers, especially TradeMark East Africa Limited or any of its donors.

## Author Contributions

**Conceptualization:** Samuel Muhula, Yvonne Opanga, Sarah Karanja, Eefje Smet, Hilke Conradi.

**Formal analysis:** Anthony Mveyange, Samuel Oji Oti.

**Methodology:** Hilke Conradi.

**Project administration:** Martha Bande, Hellen Kayiaa, Charles Leshore, David Kawai, Yvonne Opanga, Enock Marita, Sarah Karanja.

**Resources:** Samuel Muhula, Charles Leshore, David Kawai, Yvonne Opanga, Enock Marita, Sarah Karanja, Eefje Smet, Hilke Conradi.

**Supervision:** Samuel Muhula, Martha Bande, Hellen Kayiaa, Charles Leshore, David Kawai, Yvonne Opanga, Enock Marita.

**Validation:** Samuel Muhula.

**Visualization:** Anthony Mveyange, Samuel Oji Oti.

**Writing – original draft:** Samuel Muhula, Anthony Mveyange, Samuel Oji Oti, Yvonne Opanga, Eefje Smet.

**Writing – review & editing:** Samuel Muhula, Sarah Karanja, Eefje Smet, Hilke Conradi.

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
