## [Decision Letter · Decision Letter 0]

30 Oct 2020

PONE-D-20-28794

The Impact of Community Led Alternative Rite of Passage on Eradication of Female Genital Mutilation/Cutting in Kajiado County, Kenya: A quasi-experimental study

PLOS ONE

Dear Dr. Muhula,

Thank you for submitting your manuscript to PLOS ONE. After careful consideration, we feel that it has merit but does not fully meet PLOS ONE’s publication criteria as it currently stands. Therefore, we invite you to submit a revised version of the manuscript that addresses the points raised during the review process.

**The two Reviewers stated that, for the relevance and sensitivity of the topic, the manuscript deserves special attention. For this reason, the Authors are invited to a thorough review in order to make the study scientifically valid and reliable. In addition, it is essential that the Authors make all data underlying the findings in their manuscript fully available. Therefore, I invite the Authors to address all the comments raised by the Reviewers, who have provided a rich body of suggestions that the Authors can use for their review and make the manuscript suitable for publication.**

We look forward to receiving your revised manuscript.

Kind regards,

Stefano Federici, Ph.D.

Academic Editor

PLOS ONE

Journal Requirements:

3. Please include additional information regarding the interview guide or script used in the study and ensure that you have provided sufficient details that others could replicate the analyses. For instance, if you developed a guide as part of this study and it is not under a copyright more restrictive than CC-BY, please include a copy, in both the original language and English, as Supporting Information.

5.We note that [Figure(s) 1a, 1c, 2a and 2b] in your submission contain map images which may be copyrighted. All PLOS content is published under the Creative Commons Attribution License (CC BY 4.0), which means that the manuscript, images, and Supporting Information files will be freely available online, and any third party is permitted to access, download, copy, distribute, and use these materials in any way, even commercially, with proper attribution. For these reasons, we cannot publish previously copyrighted maps or satellite images created using proprietary data, such as Google software (Google Maps, Street View, and Earth). For more information, see our copyright guidelines: http://journals.plos.org/plosone/s/licenses-and-copyright.

1.    You may seek permission from the original copyright holder of Figure(s) [a, 1c, 2a and 2b] to publish the content specifically under the CC BY 4.0 license. 

Additional Editor Comments (if provided):

The two Reviewers stated that, for the relevance and sensitivity of the topic, the manuscript deserves special attention. For this reason, the Authors are invited to a thorough review in order to make the study scientifically valid and reliable. In addition, it is essential that the Authors make all data underlying the findings in their manuscript fully available. Therefore, I invite the Authors to address all the comments raised by the Reviewers, who have provided a rich body of suggestions that the Authors can use for their review and make the manuscript suitable for publication.

Reviewers' comments:

Reviewer's Responses to Questions

**Comments to the Author**

1. Is the manuscript technically sound, and do the data support the conclusions?

Reviewer #1: Partly

Reviewer #2: Yes

2. Has the statistical analysis been performed appropriately and rigorously? 

Reviewer #1: I Don't Know

Reviewer #2: I Don't Know

3. Have the authors made all data underlying the findings in their manuscript fully available?

Reviewer #1: No

Reviewer #2: Yes

4. Is the manuscript presented in an intelligible fashion and written in standard English?

Reviewer #1: Yes

Reviewer #2: Yes

5. Review Comments to the Author

Reviewer #1: Determining the impact of interventions against FGM/C, such as CLARP in this case, knows a number of methodological, practical and ethical hurdles that are difficult to overcome (e.g., Askew, 2005). The authors deserve praise for their attempt to rise to the challenge and their paper, therefore, warrants careful consideration. Their embeddedness within Amref Health Africa and the local communities with which they work have enabled the authors to meet some of these practical and ethical challenges. However, that same advantage has a drawback (social desirability, biases, etcetera). Therefore, more self-reflexivity is warranted. Also, some methodological issues remain and need further clarification.

Style: In this paper, the use of words as even, thus, given and therefore at times creates confusion, because the causal relations implied are unclear or unsubstantiated.

Background

Overall: Much attention is paid in this section to descriptive statistics surrounding FGM/C in Kenya. Although surely relevant, I’d rather see a steady build up to the problem statement. What is CLARP (rationale, history, and so on and so forth)? Why has it taken so long for decent effect studies to be conducted? What are the challenges involved? What is at stake, also for the NGO itself?

Line 28-29: It is recorded that Amref Health Africa started implementing ARPs – not CLARPs – in 2009. However, credits for the idea and design of this intervention should likely also be given to MYWO that already implemented an ARP in 1996, well before Amref Health Africa started implementing its own version. Furthermore, the authors here do not sufficiently explain and expand upon the intervention, the underlying rationale for the intervention, the applicability of the intervention in communities in which ARP does not primarily function as a rite of passage into womanhood and the added value of CL in the acronym CLARP. Isn’t CL already implied in ARP? In the end, it is not about claiming new acronyms (see also: Billig, 2013), but about actual change practices in the field.

Later in the paper (Line 398: 2.5 An overview of the CLARP model) the authors do expand upon CLARP in more detailed fashion. I would suggest to integrate this section in the introduction, as to help the reader to better understand the underlying rationale for the CLARP approach, and this research project into its effectiveness.

Line 59-90: “… FGM/C is still practised as a social norm and thus remains persistent and ubiquitous across countries.” This is kind of a casual explanation – by the use of the word ‘thus’ – of FGM/C’s persistency, but a debatable one too (e.g., Efferson et al., 2015; Hughes, 2018).

Line 77 | 140: “… paints a gloomy picture …”. In presenting numbers and figures I’d prefer more neutral language.

Line 93-94: “Finally, going through the FGM/C ordeal and the resultant psychological effects can have devastating effects on girls’ and women’s mental health.” This statement is certainly not disputed, at least not by me. However, there is a flip-side to it. Exaggerating or over-emphasizing negative (mental) health effects, or not placing these in the context of the mental health effects suffered from not-being circumcised in a community that traditionally practices it, can have devastating effects on the credibility of health workers working in these communities (e.g., Shweder, 2000). Bottom line, what do the authors precisely mean when they employ the word ‘devastating’? Words matter, and can become self-fulfilling prophecies, i.e., leading to stigmatization of circumcised girls, circumcised girls giving up on exploring for sexual pleasure/orgasms, etcetera. These dynamics may not be papered over, because these help to understand FGM/C’s persistency (see also previous point with regards to oversimplifying these dynamics under a, casually and causally used, yet purely descriptive social norm).

Line 183: “ … primarily … “ (?). Would the intervention and underlying rationale be valid in communities in which FGM/C is practiced for different reasons, or with a different functional emphasis in mind (e.g., Gruenbaum, 2001)? And, who says FGM/C functions primarily as a rite of passage, even in the communities Amref Health Africa works with. Recently, some doubts have been raised and nuances made, particularly with regards to current practices in Maasai communities (e.g., Van Bavel et al., 2017; Van Bavel, 2020).

Line 187-189: Again, the CLARP acronym might be relatively new – introduced well after 2009 even, but the intervention itself is not. The claim here ignores the contribution of other organizations that have been experimenting with these types of interventions well before 2009, and outside of Kajiado County.

Line 199: “… stands out as one of the essential community-led initiatives …”. Who says so? Since many of the authors appear to be affiliated with Amref Health Africa, a more tentative tone is likely warranted. This in order to demonstrate to readers that the authors clearly distinguish between how the intervention is marketed towards donors and other stakeholders, and that what is actually known to work – which is still very little (Droy et al., 2018). Expanding upon this more thoroughly would help justifying this research project and the chosen methods. Fairly recently World Vision – another NGO that implements ARPs in the same regions in which Amref Health Africa is active – has been criticized for the way they’ve marketed their ARP, its design, its implementation and lack of access for independent researchers to study impact (Hughes, 2018).

Line 204-206: “However, despite the declining trends in FGM/C prevalence, the extent to which the CLARP model had a role to play in such declines in Kenya, and particularly in Kajiado county remains unclear.” That is correct. So I guess readers likely expect a detailed elaboration in this introductory paragraph – after outlining what kind of practices underlie the acronym CLARP – on why this is so, since the intervention is implemented for well over a decade now. Eminent scholars on FGM/C, such as Bettina Shell-Duncan, may surely be referred to in support of the idea, but there are also those scholars with a more particular focus on ARP, or CLARP if you will, that have been critiquing the lack of evidence regarding program effectiveness that may not be ignored (e.g., Droy et al., 2018; Hughes, 2018). The authors should do all they do can to avoid being accused of cherry picking literature that favors CLARP.

Line 215-218: A firm theoretical underpinning of what is meant with experiences, stories or narratives, attitudes, perceptions and practices, and how they all relate or differ is lacking. So it is unclear of what the authors are looking for exactly.

Methods

Overall: Especially with regards to the qualitative part of the study, more information is warranted of what the researchers have done to reduce their own (confirmation) bias, and to account for social desirability in their respondents’ reactions. Most of the authors appear to be affiliated with Amref Health Africa and respondents, in some cases, may be dependent on the health care services this NGO provides for them. It is not unlikely respondents have stake in appeasing their interviewers, or in the outcomes of this project. How did the researchers account for these tendencies? What measure did they take?

Line 253: “… non-existent … ”. Taking into account the proliferation of similar interventions being implemented by a number of NGOs and lack of coordination among them, how can we be sure? Bottom line, on which sources or data is this claim based. Who has the data? The Kenyan Anti-FGM Board?

Line 344: “… purposive … “. Please specify sampling strategy. Purposive sampling is an umbrella term for a range of strategies used in qualitative research (e.g., Palinkas et al., 2015)

Line 373: Integrate in sampling paragraph.

Line 398: Move to introduction.

Discussion

I’d recommend to give more body to the discussion paragraph, for example by being more self-reflexive about the research process.

Limitations

In writing up the limitations paragraph the paper written by Ian Askew (2005) might come in handy. I think this research knows quite a number of limitations that need to be accounted for, especially in light of the final conclusion drawn, that is in definite favor of the CLARP intervention.

Conclusion

Line 758-759: “The study revealed that CLARP played a decisive role in attenuating FGM/C, CEFM and TP in Kajiado County.” Especially the word ‘decisive’ makes this a very firm statement. As yet, based on the information the authors provide, I am not convinced this firm statement is warranted. However, I would be happy to be proven wrong.

References

Askew, I. (2005). Methodological issues in measuring the impact of interventions against female genital cutting. Culture, Health & Sexuality, 7(5), 463-477.

Droy, L., Hughes, L., Lamont, M., Nguura, P., Parsitau, D., & Wamue, Ngare, G. (2018). Alternative Rites of Passage in FGM/C abandonment campaigns in Africa: A research opportunity. LIAS Working Paper Series, 1, 1-21.

Efferson, C., Vogt, S., Elhadi, A., Ahmed, H. E. F., & Fehr, E. (2015). Female genital mutilation is not a social coordination norm. Science, 349(6255), 1446-1447.

Gruenbaum, E. (2001). The female circumcision controversy: An anthropological perspective. University of Pennsylvania Press.

Hughes, L. (2018). Alternative rites of passage: Faith, rights, and performance in FGM/C abandonment campaigns in Kenya. African Studies, 77(2), 274-292.

Palinkas, L. A., Horwitz, S. M., Green, C. A., Wisdom, J. P., Duan, N., & Hoagwood, K. (2015). Purposeful sampling for qualitative data collection and analysis in mixed method implementation research. Administration and Policy in Mental Health, 42(5), 533–544.

Shweder, R. A. (2000). What about “female genital mutilation”? And why understanding culture matters in the first place. Daedalus, 129(4), 209–232.

Van Bavel, H. (2020). At the intersection of place, gender, and ethnicity: changes in female circumcision among Kenyan Maasai. Gender, Place & Culture 27(8), 1071-1092.

Van Bavel, H., Coene, G., & Leye, E. (2017). Changing practices and shifting meanings of female genital cutting among the Maasai of Arusha and Manyara regions of Tanzania. Culture, Health & Sexuality, 19(1), 1344-1359.

Reviewer #2: Overall Comment

This manuscript is a quasi-experimental study of community-based interventions on female genital mutilation/cutting (FGM/C) in one region of Kenya among the Maasai who have traditionally conducted the practice universally. The topic of the study is of importance to public health and is also a stated goal of the 2030 Sustainable Development Agenda (goal 5.3). As important as the topic is, it seems there is limited empirical evidence of the effect of FGM/C, or that the evidence is not clear-cut to warrant use of terms like “devastating”. For this, I would advise researchers to consider arguments made in this report [The Public Policy Advisory Network on Female Genital Surgeries in Africa. Seven Things to Know about Female Genital Surgeries in Africa. Hastings Cent Rep. 2012 Nov-Dec;42(6):19-27.]

Overall, the current study is well done and my assessment is that it is deserving of consideration for publication, which is my recommendation after some considerations by authors. My substantive comments follow.

Please see my full 4-page feedback attached>>>

6. PLOS authors have the option to publish the peer review history of their article (what does this mean?). If published, this will include your full peer review and any attached files.

Reviewer #1: No

Reviewer #2: **Yes: **Richard Wamai

---

## [Author Response · Author response to Decision Letter 0]

21 Jan 2021

See attachment on "Response to Reviewers". 

We have also included affiliations of all the authors on the manuscript cover page and included a disclaimer statement towards the end of the manuscript

---

## [Decision Letter · Decision Letter 1]

8 Feb 2021

PONE-D-20-28794R1

The Impact of Community Led Alternative Rite of Passage on Eradication of Female Genital Mutilation/Cutting in Kajiado County, Kenya: A quasi-experimental study

PLOS ONE

Dear Dr. Muhula,

Thank you for submitting your manuscript to PLOS ONE. After careful consideration, we feel that it has merit but does not fully meet PLOS ONE’s publication criteria as it currently stands. Therefore, we invite you to submit a revised version of the manuscript that addresses the points raised during the review process.

The reviewers appreciated the review work conducted by the authors. But still one last effort is required to make the manuscript suitable for publication. 

We look forward to receiving your revised manuscript.

Kind regards,

Stefano Federici, Ph.D.

Academic Editor

PLOS ONE

Additional Editor Comments (if provided):

The reviewers appreciated the review work conducted by the authors. But still one last effort is required to make the manuscript suitable for publication.

Reviewers' comments:

Reviewer's Responses to Questions

**Comments to the Author**

1. If the authors have adequately addressed your comments raised in a previous round of review and you feel that this manuscript is now acceptable for publication, you may indicate that here to bypass the “Comments to the Author” section, enter your conflict of interest statement in the “Confidential to Editor” section, and submit your "Accept" recommendation.

Reviewer #1: (No Response)

Reviewer #2: All comments have been addressed

2. Is the manuscript technically sound, and do the data support the conclusions?

Reviewer #1: Yes

Reviewer #2: Yes

3. Has the statistical analysis been performed appropriately and rigorously? 

Reviewer #1: I Don't Know

Reviewer #2: I Don't Know

4. Have the authors made all data underlying the findings in their manuscript fully available?

Reviewer #1: Yes

Reviewer #2: Yes

5. Is the manuscript presented in an intelligible fashion and written in standard English?

Reviewer #1: Yes

Reviewer #2: Yes

6. Review Comments to the Author

Reviewer #1: The previous recommendations, by and large, have been expertly addressed. The authors write more tentative where tentativity is required. Some further recommendations and points of attention:

81 ‘with traditional practitioners being the main perpetrators of the practice.’ Maybe specify. In contrast to trained medical professionals? What is implied here?

185 An interested layman might not know what ‘Morans’ are?

189 Same for the acronym TBA. Please clarify.

196 Same for the acronym CSO. Please clarify by using APA or journal recommended standards for employing acronyms.

359 What type of purposive sampling? Related to an earlier review, I see it is addressed, but is this a particular tried sampling strategy (e.g., homogeneous sampling, typical case sampling, extreme/deviant case sampling, maximum variation sampling, critical case sampling, total population sampling, expert sampling)?

365 ‘Barazaas’? These notions are obvious to the authors, but likely not to most readers.

373 ‘… male and female parents’. I don’t know whether this is a pleonasm, but close to it.

375 Please clarify what a ‘Moran’ is immediately when first introducing the term (line 185). This will enhance readability of the paper.

461 ‘The aspect of reporting on practices using qualitative interviews is limiting as the practice is done in secrecy.’ Sure, but this is also the case for quantitative surveys methods. What steps have been taken to deal with secrecy, possible risks for participants and the double-bind of gaining rapport and confidentiality?

465. ‘… given that experimental design was not integrated in the programme right from the start of the implementation.’ Why not?

508-514 Not sure whether this quotation is exactly on point. CLARP or ARP is not mentioned, and generally speaking, people in Eastern African communities understandably like Amref Health Africa to be active and remain in their region for their work on WASH and public health more broadly. ARP for some of the community members is a great lever to address these broader issues as well.

§3.5 Experiences and stories of CLARP versus non- CLARP beneficiaries

The title of this paragraph overpromises and underdelivers. I can imagine a lot more rigorous work underlies this paragraph, but it does not show here. It also stands in contrast to §3.6. Perhaps the authors want to stuff too much into one single paper. They might want to consider writing another paper to do justice to the underlying stories collected and experiences shared. Otherwise, I’d recommend a rewrite of this paragraph.

§3.6 Perceptions, attitudes and practices of community stakeholders about FGM/C practices

My main point of critique here is that key notions (e.g., stigma reduction, culture change) are not defined and mirrored against the existing body of literature.

3.6.2 Opportunities to improve/adapt

Please mirror these findings against existing academic literature on FGM prevention.

817 The acronyms ARP and CLARP are used interchangeably here. This is confusing to the reader. Are ARP and CLARP different interventions? Please clarify.

877 I would repeat here what was rightfully mentioned earlier … in those communities wherein FGM/C is practiced as a rite of passage into womanhood, or something along those lines.

891 ‘… act as a safe bet’ Please avoid employing metaphorical language.

894 ‘alarming rates’ Let the data speak for itself.

900-901 ‘Amref Health Africa’s noble work’ Since this is an academic paper please do not turn it into an Amref pamphlet (Amref is mentioned > 20 times), or make it appear as such. That would discredit the great work underlying this paper. That is not to say, of course, that Amref’s work is not noble. However, noble is irrelevant.

Minor points

Use of acronyms, typos, capital letters, and so on and so forth.

Reviewer #2: Dear Samuel Muhula,

I have reviewed the feedback you provided to my comments and those of the other reviewer as well as your revised manuscript. I deem you have responded to the reviewers' concerns and am happy to endorse your manuscript be considered for publication in PLOS ONE. I appreciate the time you have taken to address the detailed feedback and welcome seeing the final product once published. I also encourage you to continue this work to complete additional research to address future research questions.

Best wishes,

Professor Wamai

7. PLOS authors have the option to publish the peer review history of their article (what does this mean?). If published, this will include your full peer review and any attached files.

Reviewer #1: No

Reviewer #2: **Yes: **Richard G. Wamai

---

## [Author Response · Author response to Decision Letter 1]

20 Mar 2021

See attachment on Response to Reviewers for specific responses to the comments raised

---

## [Editor Report · Decision Letter 2]

23 Mar 2021

The Impact of Community Led Alternative Rite of Passage on Eradication of Female Genital Mutilation/Cutting in Kajiado County, Kenya: A quasi-experimental study

PONE-D-20-28794R2

Dear Dr. Muhula,

We’re pleased to inform you that your manuscript has been judged scientifically suitable for publication and will be formally accepted for publication once it meets all outstanding technical requirements.

Kind regards,

Stefano Federici, Ph.D.

Academic Editor

PLOS ONE
---

## [Editor Report · Acceptance letter]

30 Mar 2021

PONE-D-20-28794R2 

The Impact of Community Led Alternative Rite of Passage on Eradication of Female Genital Mutilation/Cutting in Kajiado County, Kenya: A quasi-experimental study 

Dear Dr. Muhula:

I'm pleased to inform you that your manuscript has been deemed suitable for publication in PLOS ONE. Congratulations! Your manuscript is now with our production department. 

Kind regards, 

on behalf of

Prof. Stefano Federici 

Academic Editor

PLOS ONE